# miRNA Dynamics for Pest Management: Implications in Insecticide Resistance

**DOI:** 10.3390/insects15040238

**Published:** 2024-03-29

**Authors:** Rashmi Manohar Mahalle, David Mota-Sanchez, Barry R. Pittendrigh, Young Ho Kim, Keon Mook Seong

**Affiliations:** 1Institute of Agricultural Sciences, Chungnam National University, Daejeon 34134, Republic of Korea; rashmimahalle27@gmail.com; 2Department of Entomology, Michigan State University, East Lansing, MI 48824, USA; motasanc@msu.edu; 3Department of Entomology, Purdue University, West Lafayette, IN 47907, USA; barrypittendrigh@gmail.com; 4Department of Ecological Science, Kyungpook National University, Sangju 37224, Republic of Korea; yhkim05@knu.ac.kr; 5Department of Applied Biology, College of Agriculture and Life Sciences, Chungnam National University, Daejeon 34134, Republic of Korea

**Keywords:** insecticide resistance, microRNA, mimic, inhibitor

## Abstract

**Simple Summary:**

MicroRNAs (miRNAs) are known to play a vital role in the evolution of insecticide resistance. This review delves into how these miRNAs regulate biological functions and their impact on insecticide resistance. It highlights the recent advancements in miRNA research, including their discovery through RNA sequencing and their regulatory roles among various insect species. It discusses the potential of miRNAs in managing insecticide resistance, focusing on the use of miRNA-based strategies. These strategies are proposed as effective and sustainable alternatives to traditional chemical pest management methods.

**Abstract:**

Utilizing chemical agents in pest management in modern agricultural practices has been the predominant approach since the advent of synthetic insecticides. However, insecticide resistance is an emerging issue, as pest populations evolve to survive exposure to chemicals that were once effective in controlling them, underlining the need for advanced and innovative approaches to managing pests. In insects, microRNAs (miRNAs) serve as key regulators of a wide range of biological functions, characterized by their dynamic expression patterns and the ability to target genes. Recent studies are increasingly attributed to the significance of miRNAs in contributing to the evolution of insecticide resistance in numerous insect species. Abundant miRNAs have been discovered in insects using RNA sequencing and transcriptome analysis and are known to play vital roles in regulation at both the transcriptional and post-transcriptional levels. Globally, there is growing research interest in the characterization and application of miRNAs, especially for their potential role in managing insecticide resistance. This review focuses on how miRNAs contribute to regulating insecticide resistance across various insect species. Furthermore, we discuss the gain and loss of functions of miRNAs and the techniques for delivering miRNAs into the insect system. The review emphasizes the application of miRNA-based strategies to studying their role in diminishing insecticide resistance, offering a more efficient and lasting approach to insect management.

## 1. Introduction

Insecticides are chemical agents employed to manage or eradicate pests that interfere in agricultural practices and human activities through the extensive application of various synthetic insecticidal compounds [1]. While they significantly contribute to crop protection and yield improvement, there is also the risk of emerging complex and evolving challenges in agriculture [2]. Although alternative control methods are likely to be introduced, chemical control will remain an integral part of modern agricultural practices for many years to come. From 1990 to 2020, the worldwide utilization of pesticides in the agricultural sector exhibited a predominantly consistent upward trend. By the end of 2020, the global consumption of pesticides reached approximately 2.7 million metric tons, reflecting a substantial growth of nearly 50 percent compared to the levels recorded in 1990 [3].

Insecticides harm the environment by contaminating soil, water, and air and disrupting ecosystems [2]. Persistent pesticide residues can accumulate in the food chain, raising concerns in terms of food safety and human health [4]. Non-target organisms like beneficial insects, birds, and aquatic life can suffer harm or death from insecticides, upsetting the natural ecological balance [5]. The repetitive or indiscriminate application of the same insecticides or those with similar modes of action can cause pest populations to become resistant. Insecticide resistance is mediated by several factors, including biological, genetic, operational, behavioral, and metabolic resistance, decreased penetration, and target site insensitivity [6,7]. The molecular mechanisms through which pests develop insecticide resistance can be categorized as target site resistance and metabolic resistance. Target site insensitivity is caused by the genetic mutations that alter the structure of specific proteins targeted by the insecticide, making it difficult for the insecticide to bind and exert its toxic effects [8]. The insensitivity of four major target sites, namely acetylcholine esterase (*AChE*), sodium channel blockers, nicotinic acetylcholine receptors (*nAChRs*), and *GABA* receptors, plays a crucial role in evolving target site resistance [9]. Metabolic resistance involves insects’ ability to metabolize or break down insecticides more efficiently before they reach and bind to the target sites, reducing their toxicity. The mechanism underlying metabolic resistance primarily relies on enzymes such as glutathione S-transferase, esterase, and cytochrome P450s, which play a vital role in the detoxification and elimination of insecticides from insect bodies [9]. The overexpression of these enzymes can neutralize insecticides before they reach their target region, making them less harmful or completely inert [10]. As a result, insects with enhanced metabolic mechanisms can survive exposure to insecticides that would be lethal to susceptible individuals.

The significance of microRNAs (miRNAs) in insects has been increasingly recognized in diverse physiological and developmental pathways through numerous research studies. The emerging evidence highlights the varied functions of miRNAs on exposure to pesticides across several organisms [11]. miRNAs induce distinct biological effects following chemical exposure and play vital roles in detoxification and defense mechanisms. Insects exposed to different insecticides exhibit significant shifts in specific miRNA expression, with some miRNAs closely tied to metabolic detoxification processes, implying their role in chemical responses [12]. These findings underscore the crucial involvement of miRNAs in organisms’ intricate reactions to pesticide exposure, with their modulation holding significant implications for insect health and survival. This review aims to shed light on how miRNAs regulate insecticide resistance and summarize recent discoveries of miRNAs conferring resistance to insecticides across various species.

## 2. Fundamentals of MicroRNAs

MicroRNAs, comprising 19–24 nucleotides, are a class of single-stranded non-coding RNAs conserved through evolution in eukaryotes, functioning as pivotal regulators of gene expression at both the transcriptional and post-transcriptional levels at their 3′ untranslated region (UTR) [13,14,15,16]. Most miRNAs typically possess a 2–8-nucleotide segment at their 5′ end, known as the seed sequence. This seed sequence binds to complementary seed match sites at the 3′ UTR of mRNAs [17]. These regulatory genes can be situated in intergenic spaces or within the intronic sequences of protein-encoding genes [18]. The interactions between miRNAs and their targets may not always suppress gene expression; in some cases, they may promote it [19] and exhibit binding to sites other than the 3′ UTR [20]. It is worth noting that specific miRNAs bind to both the 5′-UTR and the coding sequences of their target mRNA [21,22]. Intriguingly, experimental evidence has demonstrated that target sites found within the 3′ UTR of genes tend to have a more significant influence on gene regulation compared to the sites situated within the 5′-UTR or the open reading frame (ORF) in *Drosophila melanogaster* [23]. Remarkably, an individual miRNA can modulate gene expression based on the level of sequence complementarity with their targets via three significant mechanisms: the degradation of mRNA, the repression of translation, or decay of the miRNA target molecules [24,25]. These emerging insights consistently challenge the established mainstream perspectives on miRNAs and their functions, subsequently influencing the molecular and bioinformatical investigations in miRNA research.

## 3. MicroRNA Biogenesis

miRNA biogenesis involves multiple steps, initiating with the transcription of miRNA by RNA polymerase II within the cell nucleus, leading to the formation of primary miRNA (a hairpin-like structure) with a stem and a terminal loop [17]. The nuclear microprocessor assembly, consisting of the Drosha enzyme and the Pasha protein, processes primary miRNA (pri-miRNA) into a precursor miRNA (pre-miRNA), a 70-nucleotide hairpin structure [13]. Subsequently, the pre-miRNA undergoes further processing and is shuttled from the nucleus to the cytoplasm via the Exportin-5 protein (Figure 1).

In the cytoplasm, Dicer, which is an RNaseIII enzyme, carries out a series of processing steps, leading to the formation of a ~22-nucleotide double-stranded RNA duplex known as the miRNA–miRNA* duplex [17]. This duplex gives rise to two strands, the miRNA and miRNA* strands, commonly referred to as the guide and passenger strands, respectively. Subsequently, the mature miRNA strand is selectively integrated into the RNA-induced silencing complex (RISC), where the Argonaute protein functions as the key component. During the loading phase, the passenger strand is degraded, leaving only the guide strand to steer the RISC (Figure 1). This guide strand empowers the RISC to bind to the mRNA by matching the target sequence, ultimately resulting in the suppression of gene expression [15,16].

## 4. Insect miRNAs

Since the discovery of the first miRNA in *Caenorhabditis elegans* in the 1990s, miRNAs have been identified in several organisms through molecular cloning techniques and computational predictions. Previous research has revealed that miRNAs constitute about 1–5% of all animal genes and regulate approximately 20–30% of genes [26]. For insects, the discovery of miRNA was initially reported in *D. melanogaster* [27]. Subsequently, extensive research efforts have resulted in the discovery of a considerable number of insect miRNAs, and these sequences have been deposited into the miRBase, comprising a total of 4330 miRNAs (available at http://www.mirbase.org, accessed on 27 March 2024). The InsectBase version 2.0 database reveals a huge discovery of 112,162 miRNAs across 807 insect species [28,29]. As sequencing technologies continue to advance, ongoing investigations have not only uncovered new miRNAs but also revised previously known ones. In insects, miRNAs play pivotal roles in an array of pathways and biological processes, including embryogenesis, molting, metamorphosis, wing development, immunity, reproduction, host–pathogen interactions, and insecticide resistance [30,31,32]. These regulatory functions shed light on the intricate processes underlying insect biology and have potential implications for pest management and control strategies.

Studies have shown that miRNAs frequently correlate negatively with target gene expression [11]. For instance, *Spodoptera frugiperda* larvae exhibited a down-regulation in the expression of miR-190-5p, with an up-regulation in the target gene *CYP6K2*, following exposure to chlorantraniliprole [33]. miR-1 is an extensively investigated conserved miRNA in mammals and insects and plays a crucial role in key physiological processes, including growth and development. It has been linked to various critical functions across different insect species, including the preservation of egg structure integrity through targeting a gene responsible for a vitelline membrane protein in *Bombyx mori*, involvement in immune responses in *D. melanogaster*, and differential expression in response to pathogen infection in *Aedes aegypti* [34,35,36]. miR-1 is also reported as a conserved miRNA targeting the glutathione S-transferase gene. A recent study revealed that the target site of miR-1-3p might be highly conserved in *Tetranychus cinnabarinus* [37] and *S. frugiperda* [32], strongly suggesting a miRNA-mediated role in insecticide resistance. Typically, a single miRNA can target several genes, and conversely, a single gene can be regulated by numerous miRNAs [38]. For example, miR-278 is known to regulate a range of functions by targeting genes such as the pyrethroid-resistance-related gene (*CYP6AG11*) and insulin-related peptide binding protein 2 (*IBP2*). This regulation modulated energy balance, insecticide resistance, and immune responses across species, including *D*. *melanogaster*, *B*. *mori*, and *Culex pipiens pallens* [38,39]. Recent studies have highlighted the significance of detoxification enzyme genes in insect immune defenses and their pivotal role in the development of insecticide resistance across various insect species [37]. Despite the extensive diversity of insect species and the complexity of their regulatory networks, an understanding of the specific dynamics of miRNA–target gene interactions across different species remains largely unknown.

## 5. Approaches for Investigating miRNAs in Insecticide Resistance

The proposed approach to studying miRNAs in insecticide resistance involves two key steps: 1. Profiling miRNA expressions using sequencing technologies and comparing them to identify differentially expressed miRNAs in resistant strains. 2. Inducible gene expression post-insecticide treatment. miRNA profiling begins with isolating RNA from both insecticide-resistant and susceptible insect strains to construct a small RNA library using high-throughput sequencing technologies [40]. The data obtained from each library are aligned with reference genomes or known miRNA databases, which systematically identifies unique miRNA sequences from resistant and susceptible strains. Bioinformatical analysis identifies the differentially expressed miRNAs (either up-regulated or down-regulated) in the resistant strains compared to the susceptible ones [41]. Furthermore, in silico prediction is extensively utilized to identify potential targets linked to these differentially expressed miRNAs. This involves investigating their roles in metabolic pathways, detoxification processes, and other cellular mechanisms related to resistance [42].

The approach of inducible gene expression post-insecticide treatment in insects involves studying how the expression of miRNAs and their target genes change in response to insecticide exposure. Insects from a susceptible strain are exposed to a sub-lethal dose of an insecticide for specific time intervals under controlled laboratory conditions. Insects are collected at different time points following insecticide exposure to confirm the expression patterns of the differentially expressed miRNAs discovered using small RNA sequencing [41]. RNA (including miRNA) is extracted from the collected samples to quantify the expression patterns of these differentially expressed miRNAs and their predicted target genes in response to insecticide exposure compared to untreated control insects using RT-qPCR. Additionally, specific tissues (like the head, gut, cuticle, malpighian tubule, or fat body) may be targeted for analyzing miRNA and gene expression to study the spatiotemporal dynamics and miRNA functions in insects [42]. Overall, the results might indicate that miRNAs modulate resistance to insecticides by targeting genes that are presumed to be involved in the detoxification of insecticides [40,41,42,43]. This approach is crucial to unraveling the molecular mechanisms of insecticide resistance and significant for developing more effective pest management strategies.

## 6. miRNA-Mediated Insecticide Resistance in Different Arthropod Species

Comprehending the modulatory functions of miRNAs in resistance to insecticides offers critical perspectives for developing potent pest management methods. Recent investigations into insect miRNAs have highlighted their profound influence on core cellular activities and gene expression dynamics. Increasingly, studies are shedding light on miRNAs’ involvement in fostering insecticide resistance in arthropods. This review explores the significance of miRNA identification in arthropods and emphasizes their role in resistance development (Table 1).

### 6.1. Arachnida

Zhang et al. (2016) [44] discovered a total of 139 miRNAs (75 known and 64 novel) in fenpropathrin-resistant (TR) and susceptible strains (TS) of *T. cinnabarinus*. Out of these, 12 miRNAs exhibited significant differences in their expression levels. Functional studies suggested that these miRNAs likely target the genes associated with detoxification enzymes, including 7 from the cytochrome P450 family, 7 glutathione S-transferases, and 6 carboxyl/choline esterases, indicating their probable role in resistance when exposed to fenpropathrin. Additionally, research on *T. cinnabarinus* presented empirical data, highlighting the potential role of miR-1-3p in resistance to cyflumetofen. This particular miRNA was observed to target a gene responsible for encoding a *GST* (glutathione S-transferase) enzyme, suggesting its involvement in the resistance pathway [37].

### 6.2. Diptera

In the initial study of miRNA expression in deltamethrin-sensitive and resistant strains of *Culex pipiens*, it was confirmed that cpp-miR-71 interacts with the 3′UTR of *CYP325BG3*, providing compelling evidence of its role in mosquito pyrethroid resistance [45]. miR-278-3p was found to be involved in regulating pyrethroid resistance in *C. pipiens* pallens [39]. Tian et al. (2016) [46] confirmed that miR-285 control pyrethroid resistance in *C. pipiens* by targeting *CYP6N23*. The involvement of miR-932 in pyrethroid resistance in *C. pipiens pallens* regulated a target gene, *CpCPR5* [47]. One study offered pioneering evidence showing that the miR-2~13~71 cluster had significantly decreased expression, which played a crucial role in conferring deltamethrin resistance by controlling the expression of the target genes *CYP325BG3* and *CYP9J35* in deltamethrin-resistant (DR) mosquitoes [48]. Ma et al. (2017) [49] demonstrated that the increased expression of miR-92a in a deltamethrin-resistant strain of *C. pipiens pallens* could potentially target the mosquito cuticle gene *CpCPR4a*. This regulation of *CpCPR4* was further corroborated using dual-luciferase reporter assays that showed specific binding in the 3′ UTR of the gene. The pivotal role of miR-13664 and its interaction with *CpCYP314A1* were reported in managing deltamethrin resistance in *C. pipiens pallens* [50]. Through a combination of bioinformatics, luciferase assays, and mimic/inhibitor injections, *CpCYP314A1* was established as a definitive target gene of miR-13664, shedding light on new avenues in understanding mosquito resistance mechanisms. In another study, the first evidence of L1014F knockdown resistance gene (*Kdr*) mutations in *C. pipiens pallens* presented an inverse relationship between miR-285 and its presumed target gene, *CYP6Cp1* [51].

Another study provided evidence that miR-279-3p could suppress the expression of *CYP325BB1*. This suppression resulted in diminished deltamethrin resistance and a consequent rise in mosquito mortality [52]. miR-4448 has been identified to participate in the regulation of *CYP4H31* gene that contributes to deltamethrin resistance in *C. pipiens* [53]. A miRNA identified as miR-310c has a regulatory effect on the development and function of organs involved in chemical perception by modulating *Esg* gene expression in *D. melanogaster* [54]. Seong et al. (2018) [40] investigated the miRNA expression in *D. melanogaster* strains, the DDT-resistant 91-R, and the susceptible Canton-S. They analyzed the miRNA profiles post-DDT exposure. Remarkably, a series of miRNAs, such as miR-92a/310/311/312/313-3p, emerged as likely agents in DDT detoxification by modulating distinct cytochrome P450s. Furthermore, a reduction in the miR-313-3p, miR-312-3p, and miR-311-3p expression levels, paired with a rise in the expression of predicted target genes such as *Cyp4ae1*, *Cyp4s3*, *Cyp6a8*, *Cyp6t3*, *Cyp6v1*, *Cyp6g2*, *Cyp18a1*, *Cyp303a1*, *Cyp49a1*, *Cyp313a4*, and *Cyp309a2* in the 91-R strain, suggests a miRNA-driven post-transcriptional modulation associated with insecticide resistance [12].

### 6.3. Hemiptera

In a study examining the miRNA profile of *Myzus persicae nicotianae*, 22 miRNAs were predicted to target *CYP6CY3*. Notably, let-7 and miR-100 emerged as key regulators of the *CYP6CY3* gene associated with nicotine tolerance. Interestingly, a reduction in *CYP6CY3* levels resulted in a significant increase in the homologous *CYP6CY4* gene, implying a potential compensatory relationship between the two [55]. To decipher the post-transcriptional regulation mechanism of the *ACC* gene (acetyl-CoA carboxylase) in *Aphis gossypii*, a prediction using its 66 known miRNAs identified miR-276 and miR-3016 as potential *ACC* regulators, with their abundance showing inverse correlations with *ACC* transcript levels [56]. Ma et al. (2019) [57] unveiled the regulatory role of miR-4133-3p in the post-transcriptional expression of *CYP4CJ1*. Additionally, they identified a promoter region that responded to certain plant allelochemicals like gossypol and tannic acid in *A. gossypii*. The effects of imidacloprid on *Sitobion miscanthi* were examined by identifying 265 miRNAs, out of which 23 were up-regulated and 54 were down-regulated. Further, smi-miR-278 and miR-263b were discerned to regulate *nAChRα1A* and smi-miR-316 was involved in regulating *CYP4CJ6*, emphasizing their pivotal roles in *S. miscanthi*’s metabolism and imidacloprid resistance [58,59].

Three miRNAs (PC-5p-3096_674, PC-3p-2522_840, and PC-3p-446_6601) were reported to modulate resistance to triflumezopyrim by regulating the detoxification genes, including *ABCA3*, *UGT386F1*, *CYP6FL1*, and *GSTD2* in the small brown planthopper (SBPH) [60]. Another study focused on *Nilaparvata lugens* associated the *ABC* transporter *NlABCG3* with resistance to nitenpyram and clothianidin. A significant overexpression of *NlABCG3* was noted in resistant populations compared to susceptible ones. Furthermore, miRNA novel_268 was identified as a regulator of *NlABCG3* expression, playing a role in the mechanisms of resistance to these insecticides [61]. Extending this research, a comprehensive miRNA screen was conducted to pinpoint miRNA-based targets for resistance management against nitenpyram in *N. lugens*, and 220 miRNAs were identified, with 57 differentially expressed between the resistant and susceptible strains. Notably, novel_85 and novel_191 miRNAs were identified as potential regulators of *CYP6ER1* and *CarE1*, respectively, with subsequent tests confirming their regulatory role [62].

### 6.4. Coleoptera

Numerous investigations have been conducted to understand the roles of miRNAs in various coleopteran insects, including *Tribolium castaneum* [63], *Trachyderma hispida* [64] *Holotrichia parallela* [65], *Diabrotica virgifera* [66], etc. However, studies focusing on their involvement in insecticide resistance are still limited. In a study, the expression profiles of miRNAs, particularly miR-282 and miR-989, suggested their functional role in targeting cytochrome P450 expression in imidacloprid-treated *Leptinotarsa decemlineata* [67].

### 6.5. Lepidoptera

The overexpression of *PxRyR* in *Plutella xylostella,* modulated by miR-7a and miR-8519, is implicated in resistance to chlorantraniliprole [68]. Etebari et al. (2018) [11] highlighted miRNAs such as miR-14b-5p, let-7-5p, and miR-2b-3p that are linked with insecticide resistance in *P. xylostella*. Exposure to chlorantraniliprole increases *CYP9F2* and *CYP307a1* expression, impeding larval detoxification pathways. Several conserved miRNAs, including Let-7, miR-1, miR-184, miR-10, and miR-275, exhibited high expression post-*Bacillus thuringiensis* exposure in *P. xylostella* [69]. Investigation of the regulation of fufenozide resistance in *P. xylostella* confirmed that miR-189942 binds to the 3′-UTR of the *PxEcR-B* gene, which subsequently down-regulated *PxEcR-B* expression, altering insects’ sensitivity to fufenozide. However, it had no impact on their sensitivity to chlorantraniliprole, as the latter does not target *PxEcR-B* [70]. Zhu et al. (2021) [71] revealed that *lnc-GSTu1-AS* enhances chlorantraniliprole metabolism by hiding miR-8525-5p’s binding site in the 3′ UTR of *GSTu1*, which, in turn, increases the resistance of *P. xylostella*. A study detailed 12 differentially expressed miRNAs between the Cry1S1000 and G88 strains of *P. xylostella*. Specifically, novel-miR-240 connected with targets *Px017590* and *Px007885* was predicted to play a crucial role in mediating Bt resistance [72]. The differential expression of the gene *PxABCG20* and *PxJHE* was associated with resistance to *Cry1Ac* toxin, likely to be regulated by novel-miR-310 and novel miR-108, miR-234, respectively, in *P. xylostella* [73,74]. 

Injection of miR-998-3p mimic was found to down-regulate *ABCC2* expression across *Helicoverpa armigera*, *S. exigua*, and *P. xylostella*, providing insights into the mechanisms of *Cry1Ac* resistance in lepidopteran pests [75]. A recent study identified 30 differentially regulated miRNAs after tetraniliprole treatment, predicted the functional roles of the target genes affected by these differentially expressed miRNAs, and verified the impact of miR-278-5p’s association with tetraniliprole susceptibility in *S. frugiperda* [42]. With RT-qPCR validation of miR-13b-3p, miR-10483-5p, miR-10485-5p, and miR-278-5p, they were observed to undergo significant down-regulation following exposure to three distinct insecticides. Interestingly, these overexpressed miRNAs enhanced the mortality rate of *S. frugiperda* when exposed to treatments with cyantraniliprole, emamectin benzoate, and spinetoram. These findings indicate that these differentially expressed miRNAs might act as potential regulators in mediating resistance in *S. frugiperda* [32]. Zhang et al. (2022) [33] identified a miRNA-190-5p binding site in the 3′ UTR of *CYP6K2*, suppressing its expression and thereby affecting resistance to chlorantraniliprole in *S. frugiperda*. A total of 10 co-differentially expressed miRNAs were predicted to regulate indoxacarb resistance through indoxacarb-resistance-related genes in *S. litura* [76].

**Table 1 insects-15-00238-t001:** Reports on miRNA-mediated resistance to insecticides in pests of agricultural importance.

Order	Insect Species	Life Stage	miRNA	Target Gene(s)	Strategy	Insecticide	Mode of Action	Reference
Arachnida	*Tetranychus* *cinnabarinus*	Adult	tci-miR-281-5p, tci-miR-281-3p tci-miR-92-3p	*TuCCE-29*, *TuGSTm01*, *CYP392E9*, *TuGSTd01*		Fenpropathrin	Sodium channel modulators	[44]
			tci-miR-1-3p	*TCGSTM4*	Treated leaf-discs feeding	Cyflumetofen	Mitochondrial complex II electron transport inhibitors	[37]
Diptera	*Culex pipiens*	Larval	miRNA-71	*CYP325BG3*	Microinjection	Deltamethrin	Sodium channel modulators	[45]
			miRNA-278-3p	*CYP6AG11*				[39]
			miRNA-285	*CYP6N23*				[46]
			miRNA-932	*CpCPR5*				[47]
			miR-2/13	*CYP9J35*				[48]
			miRNA-92a	*CpCPR4*				[49]
			miR-13664	*CYP314A1*				[50]
			miR-285, miR-2, miR-13, miR-71, miR-278-3p	*CYP6N23*, *CYP6Cp1*, *CYP9J35*, *CYP325BG*, *CYP6AG11*	-			[51]
			miRNA-279-3p	*CYP325BB1*	Microinjection			[52]
			miRNA-4448	*CYP4H31*				[53]
	*Drosophila* *melanogaster*	Adult	miR-310~313	*Esg* gene	Volatilized	Nicotine	Nicotinic acetylcholine receptor competitive modulators	[54]
				*Cyp6g1*, *Cyp6g2*	Microinjection	Dichlorodiphenyl- trichloroethane	Sodium channel modulators	[31]
Hemiptera	*Myzus persicae*		let-7, miR-100	*CYP6CY3*	Artificial diet (sterile sucrose)	Nicotine	Nicotinic acetylcholine receptor competitive modulators	[55]
	*Aphis gossypii*	Third instar nymphs	miRNA-276, miRNA-3016	*Acetyl-CoA* *carboxylase*		Spirotetramat	Inhibitors of acetyl-CoA carboxylase	[56]
			miRNA-4133-3p	*CYP4CJ1*		Gossypol and tannic acid	-	[57]
	*Sitobion* *miscanthi*		smi-miR-316, smi-miR-278	*CYP4CJ6*, *nAChRα1A*		Imidacloprid	Nicotinic acetylcholine receptor competitive modulators	[58]
			miR-263b	*nAChRβ1*				[59]
	*Laodelphax* *striatellus*		PC-3p-2522_840, PC-3p-446_6601 PC-3p-3096_674	*CYP6FL1*, *GSTD2*, *UGT386F1*, *ABCA3*		Triflumezopyrim		[60]
	*Nilaparvata lugens*	Fourth instar nymph	miRNA novel_268	*NlABCG3*	Microinjection	Nitenpyram and clothianidin		[61]
			novel_85, novel_191	*CYP6ER1*, *CarE1*		Nitenpyram		[62]
Coleoptera	*Leptinotarsa* *decemlineata*		miR-282, miR-989	No validated targets	-	Imidacloprid		[67]
Lepidoptera	*Plutella* *xylostella*	Third instar larva	Pxy-miR-7a, Pxy-miR-8519	*PxRyR*		Chlorantraniliprole	Ryanodine receptor modulators	[68]
			miR-2b-3p, miR-14b-5p	*CYP9F2*, *CYP307a1*	Artificial diet feeding			[11]
			miRNA-2b-3p	*Trypsin*	Luciferase reporter assay	*Bacillus* *thuringiensis* (Bt Cry1Ac)	-	[69]
			miR-1, Let-7, miR-275, miR-184	No validated targets				
			miR-189942	*PxEcR-B*	Microinjection	Fufenozide		[70]
			miR-8525-5p	*GSTu1*		Chlorantraniliprole	Ryanodine receptor modulators	[71]
			novel-miR-240	*Px017590*, *Px007885*	Luciferase reporter assay	*Bacillus* *thuringiensis* (Bt Cry1Ac)	-	[72]
			Novel-miR-310	*PxABCG20*	Microinjection			[73]
			Novel miR-108, miR-234	*PxJHE*				[74]
	*Helicoverpa* *armigera* *Spodoptera* *exigua* *Plutella xylostella*		miR-998–3p	*ABCC2*				[75]
	*Spodoptera* *frugiperda*		miR-190-5p	*CYP6K2*		Chlorantraniliprole	Ryanodine receptor modulators	[33]
			miR-278-5p, miR-13b-3p, miR-10485-5p, miR-10483-5p	No validated targets		Cyantraniliprole		[32]
						Spinetoram	-	
						Emamectin benzoate	Glutamate-gated chloride channel allosteric modulators	
			miR-278-5p			Tetraniliprole	Ryanodine receptor modulators	[42]

## 7. Validation of Identified microRNAs

### 7.1. Gain- and Loss-of-Function Analyses

Gain- and loss-of-function analyses represent a cutting-edge method for elucidating the specific functions of individual microRNAs [77]. In insects, the gain and loss of function involves manipulating the miRNA levels to study their effects (Figure 2). A miRNA identified as critical in gain-of-function experiments can be further validated using loss-of-function experiments, and vice versa. Using both approaches, endogenous miRNA concentrations can be altered to investigate the impact on protein expression [78]. The loss or absence of specific miRNAs might result in minor or no observable phenotypic changes. Under such circumstances, a gain-of-function approach is often employed using exogenous miRNA mimics (agomirs) to observe the hypothetical enhancement of the regulatory role of miRNAs on their target genes and the resulting phenotypic changes in insects [19]. Chemically synthesized double-stranded RNAs called miRNA mimics have an identical sequence to that of mature miRNAs [17]. When introduced into cells, these mimics act like endogenous miRNAs, specifically binding to their target gene by post-transcriptional repression, primarily through translation gene inhibition [79] (Figure 2A). Additionally, they can be used to enhance the expression of these miRNAs within specific tissues, providing an understanding of miRNA’s biological roles in living organisms [80].

Loss-of-function experiments are crucial to investigating miRNAs that are highly expressed naturally and presumed to have a significant regulatory impact on their target genes. miRNA inhibitors, commonly known as antagomirs, silence or knock down endogenous miRNA function and demonstrate the impact of reduced miRNA activity on gene expression [81]. Antagomirs are synthetic RNA molecules designed to specifically bind and inhibit the function of specific miRNAs (both single miRNAs and entire miRNA seed families), thereby allowing the investigation of the roles of specific miRNAs [82] (Figure 2A). However, loss-of-function studies on miRNAs are prone to affecting RNA species other than the miRNA target [83].

Cultured insect cells can be transfected with miRNA mimics or inhibitors to study the functions of specific miRNAs, how they influence gene expression, and their impacts on cellular processes [17]. The Gal4/UAS system has been employed to study microRNA gain- or loss-of-function experiments in specific tissues and at specific developmental stages. A novel transgenic approach known as a “microRNA sponge (miR-SP)” has been successfully employed in *D*. *melanogaster* [17]. This innovative method enables the investigation of specific microRNA functions in various biological processes, including insecticide resistance, with exceptional precision and exhibits the potential for miRNA-mediated control strategies.

### 7.2. miRNA–Target Gene Interactions

The investigation of the interactions between miRNAs and their target sites on mRNAs is crucial for elucidating the functional roles of miRNAs. Firefly luciferase is widely used as a reporter to evaluate the miRNA-mediated post-transcriptional regulation of target genes in cells. This involves constructing a luciferase reporter gene construct that incorporates the miRNA target sequence, typically found in the 3′-UTR of the target mRNA, thereby allowing for the direct measurement of miRNA-mediated gene expression [84]. Recent studies using a luciferase-based reporter system in *Drosophila* S2 cells have indicated that miRNA action primarily initiates through translational repression, followed by mRNA deadenylation and subsequent decay [17,85]. The results from the luciferase assay indicated that the miR-2b-3p mimic notably reduced the expression of the target gene Trypsin, suggesting its significant involvement in the defense mechanism of *P*. *xylostella* against *B*. *thuringiensis* infection [69]. In another study, the interaction of novel-miR-240 with its target genes *Px017590* and *Px007885* was confirmed using dual-luciferase reporter assays, demonstrating an inverse relationship and probable roles in mediating *Bt* resistance [72].

## 8. Insights into miRNA Mimics and Inhibitors in Insect Research

Mimics and inhibitors of miRNAs are synthetic molecules used to manipulate the activity of miRNAs in cells. The study of insect miRNA function involves the introduction of miRNA mimics and inhibitors into insects’ systems via oral ingestion or microinjection. This research investigates how these artificially designed molecules can alter miRNA levels, thereby influencing the expression of genes that may play a role in the detoxification pathways of insects. Initially, the insects under study are subjected to either mimic/inhibitor molecules corresponding to specific miRNAs, including a negative control mimic/inhibitor for comparison. Subsequently, RNA is extracted from the insects after a specified time interval, and the expression of both the miRNA and its target gene is analyzed using qRT-PCR [17]. This approach enables us to determine whether miRNA mimics and inhibitors have successfully altered the miRNA and target gene expression levels and thus helps in elucidating the mechanisms through which miRNAs regulate these specific genes [11,42].

Different techniques for delivering mimics/inhibitors are developed based on insect species and primarily categorized into microinjection and oral ingestion (Figure 2B). The larval, nymphal, or adult stage of insects is often chosen for studies on miRNA related to insecticide resistance (Table 1) because insects tend to be more receptive to external stimuli and generally exhibit higher feeding rates during these phases [29].

### 8.1. Microinjection

Microinjection enables the delivery of mimics or inhibitors directly into target tissues or the hemolymph, circumventing physical barriers like the integument or gut epithelium. The first microinjection experiment was performed on *D. melanogaster* [86], and subsequent investigations have been conducted on numerous other insect species, including *C. pipiens pallens* [39,46,47,48,49], *Laodelphax striatellus* [60], *N. lugens* [61,62], *P. xylostella* [70,71], and *S. frugiperda* [32,33,42]. The microinjection technique offers the precise and direct introduction of a defined quantity and concentration of miRNA (mimics or inhibitors) into the desired location. Nonetheless, it requires substantial labor, specialized knowledge, and equipment rendering it unsuitable for field-based applications [87].

The microinjection-induced overexpression of miR-278-3p resulted in a substantial decrease in the survival rate of *C. pipiens pallens*, concomitantly reducing the expression levels of *CYP6AG11* and regulating pyrethroid resistance [39]. Down-regulating *CYP6N23* using miR-285 mimic microinjection reduced the mortality of *C. pipiens pallens*, providing insights into resistance surveillance and control [46]. Notably, increased expression of miR-932 resulted in the down-regulation of *CpCPR5*, and the subsequent knockdown of *CpCPR5* decreased mosquitos’ sensitivity to deltamethrin in the DS strain of *C. pipiens pallens* [47]. Microinjection of the miR-92a inhibitor up-regulated *CpCPR4* expression, increasing susceptibility in the DR strain of *C. pipiens pallens* [49]. Furthermore, manipulating the expression of novel_85 and novel_191 using miRNA mimics through injection significantly increased the susceptibility of *N. lugens* to nitenpyram [62]. The injection of an agomir (mimic) or antagomir (inhibitor) to modulate miR-7a and miR-8519 had an impact on the levels of *PxRyR* expression, subsequently affecting the susceptibility of *P. xylostella* to chlorantraniliprole [68]. Similarly, microinjecting a novel miR-310 agomir resulted in reduced expression of *PxABCG20*, which subsequently resulted in enhanced tolerance to *Cry1Ac* protoxin within the susceptible strain of *P. xylostella* [73]. Injection of the miR-998-3p agomir significantly reduced the abundance of *ABCC2*, accompanied by an increased tolerance to the *Cry1Ac* toxin in lepidopteran larvae [75]. A miRNA-190-5p agomir injection significantly reduced *CYP6K2*’s abundance, improving chlorantraniliprole tolerance in *S. frugiperda* larvae [33].

### 8.2. Oral Ingestion

Oral ingestion involves treating an artificial diet with mimic or inhibitor miRNA using various methods, such as immersion, surface contamination, percolation, or soaking. With this approach, the miRNA molecules are incorporated into the diet to enable ingestion by the target insects. Prior studies have demonstrated antagomirs and agomirs being introduced into the artificial diet of many insect species, including *H. armigera* [88], *A. gossypii* [56], *P. xylostella* [11], *S. miscanthi* [58], *Bactrocera dorsalis* [89], etc. The advantages of oral ingestion are that it resembles insects’ natural feeding process, allowing insects to consume miRNA as part of their regular diet, and its easy application. Unlike microinjection, it does not require physical intrusion into an insect’s body, thus reducing stress and potential disruptions. However, this method may have challenges related to the miRNA’s stability during processing or storage of the treated diet and variations in the dose and uptake efficiency among different insect species. Another significant drawback is that miRNA mimics delivered via feeding can be degraded by nucleases in insects’ digestive tract and gut, reducing their efficacy [80,90].

Modulating miR-276 and miR-3016 by adding mimics to *A. gossypii*’s diet decreased its *ACC* levels and spirotetramat tolerance, confirming their role in regulating resistance [56]. Upon introducing the corresponding antagomir/inhibitor molecules into an artificial diet, the expression of specific miRNAs (smi-miR-316, smi-miR-1000, and smi-miR-iab-4) significantly changed the susceptibility of *S. miscanthi* to imidacloprid [58]. Over-supplementation of the miRNAs PC-3p-2522_840, PC-3p-446_6601, and PC-5p-3096_674 in an artificial diet knocked down the target genes (*CYP6FL1*, *GSTD2*, *UGT386F1*, and *ABCA3*) and further increased *L. striatellus* (SBPH)’s mortality when it was exposed to triflumezopyrim [60]. Additionally, applying miR-2b-3p and miR-14b-5p mimics effectively reduced the expression of *CYP9F2* and *CYP307a1*, respectively, in *P. xylostella* cell lines. Further, the incorporation of mimics into their diet significantly increased mortality in deltamethrin-resistant larvae upon exposure to deltamethrin. These findings indicated that miR-2b-3p could down-regulate the *CYP9F2* transcript levels in *P. xylostella*, thereby inhibiting larval detoxification pathways [11].

## 9. Conclusions and Future Implications in Pest Management

The rapid development of high-throughput small RNA sequencing technology and the discovery of miRNAs in insects as key players in regulating gene expression transcriptionally have significantly increased our understanding of various biological processes, including insecticide resistance and detoxification in insects [17]. Detoxification genes, identified as targets, are regulated by specific miRNAs. These interventions can directly impact resistance traits in pest populations by either up-regulating the miRNAs that suppress resistance genes or down-regulating them to enhance them [91]. Despite their potential, to date, only a few studies have explored the use of miRNAs for managing insecticide resistance, yet the specific functions of most of these presumed miRNAs remain to be elucidated. MiRNA profiling experiments to identify specific miRNAs that exhibit altered expression levels on exposure to chemicals are commonly identified as the primary method for understanding pesticide resistance in insects [91]. Other major challenges include the inadequacy of genome annotations, the unavailability of exhaustive 3′ UTR databases, and the absence of advanced miRNA target prediction tools, collectively hindering the precise identification of miRNA targets in insect species [17].

Researchers can design tailored interventions, such as mimics or inhibitors of particular miRNAs that disrupt resistance development and offer emerging opportunities for pest management [17]. RNA interference (RNAi) is a powerful biological mechanism triggered by double-stranded RNA (dsRNA), leading to sequence-specific gene silencing by directing the degradation of targeted mRNA, effectively inhibiting the translation of proteins [92]. Another potential strategy for insect pest management envisages the utilization of miRNAs in trans-kingdom RNA interference (TK-RNAi), wherein artificial miRNAs (amiRNAs) are designed to transform through the diet across kingdoms from one species to another species, offering a novel approach to pest management. This technology was used to enhance the expression of amiRNA-319a-*HaEcR* in tomatoes, which resulted in resistance towards the ecdysone receptor (*EcR*) gene of *H*. *armigera* [93]. Recently, a plasmid vector for RNAi with an amiRNA sequence aimed at the acetylcholine esterase 1 (*Ace 1*) gene was successfully introduced into tomato varieties, which developed resistance to the aphids *M. persicae* [94].

Despite these advancements, directly applying formulated synthetic miRNAs through spraying might emerge as the most promising strategy against insect pests in agricultural fields [91,95]. This approach has been proven effective in a few studies among herbivorous and piercing/sucking insects; however, it significantly broadens the scope of utilizing miRNAs to manage a wider range of insect pests [91]. While attaining stability and efficient in vivo delivery of miRNAs/RNAi remains a challenging endeavor, advancements in this area offer a deeper understanding of the functional role of miRNAs in resistance mechanisms. miRNAs’ stability can be enhanced through the exploration of RNA chemical modifications, including modifications to the ribose 2′-hydroxyl group or the phosphorothioate backbone and the use of locked or unlocked nucleic acids to create synthetic miRNAs [96]. Several studies have evaluated the potency of different miRNA delivery methods, including genome editing techniques (CRISPR-based editing) and the use of viral vectors and non-viral vectors (polymer-based and lipid-based vectors, nanoparticles, and cell-derived membrane vesicles) [92].

miRNA-based strategies enable the specific targeting of genes associated with insecticide resistance. It is possible to break the resistance mechanisms, making pests once again susceptible to existing insecticides, thereby offering the long-term effectiveness of these chemical controls. The development and implementation of miRNA-based strategies facilitate the design, optimization, and delivery of miRNA-based solutions tailored to specific requirements [17]. This approach can reduce the reliance on chemical insecticides, decreasing the risk of resistance development. miRNA-based resistance regulation can also be combined with other integrated pest management (IPM) strategies to create a multifaceted approach to pest control [7]. This is due to the molecular mechanism of action of miRNAs and the possibility of targeting multiple genes or pathways simultaneously [95]. Developing genetically engineered insect-resistant crop varieties significantly expands the range of target genes available for pest management [97]. This approach not only protects crops but also reduces the need for external pesticide applications. Emerging evidence suggests that miRNAs play a pivotal role in regulating biological mechanisms during insect–plant interactions, eventually resulting in the development of resistance to pests [95]. Unlike chemical insecticides, miRNA interventions can be designed to target only specific pest species, reducing unintended impacts on beneficial insects and the surrounding ecosystem [98]. Ongoing research continues to discover new target genes and mechanisms, thereby expanding the potential of miRNA-based interventions for sustainable and innovative pest management solutions.

Our review article provides an overview of the identified miRNAs and their predicted target genes across various insect species, revealing their possible roles in regulating resistance to several insecticides. miRNAs have emerged as pivotal biomarkers for detecting specific targets and enhancing our understanding of resistance mechanisms. Despite their promising results, the application of miRNA-based approaches is in its initial phases, and further research is required to fully explore and utilize the potential of miRNAs in regulating insecticide resistance.

## Figures and Tables

**Figure 1 insects-15-00238-f001:**
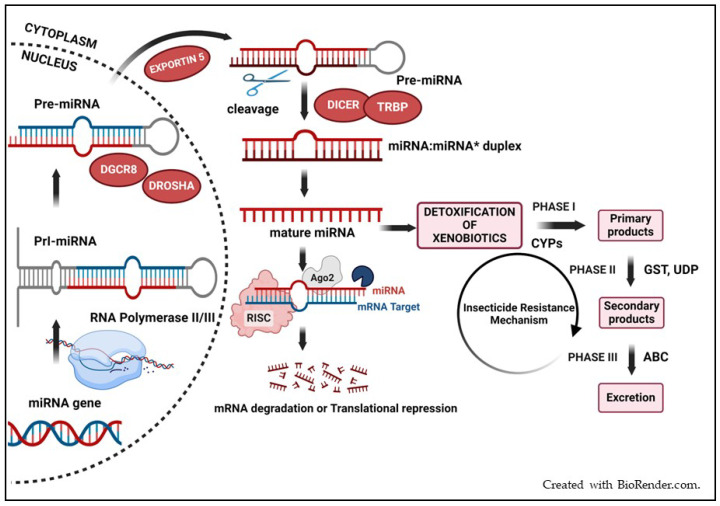
Schematic diagram of miRNA biogenesis and regulation of gene expression.

**Figure 2 insects-15-00238-f002:**
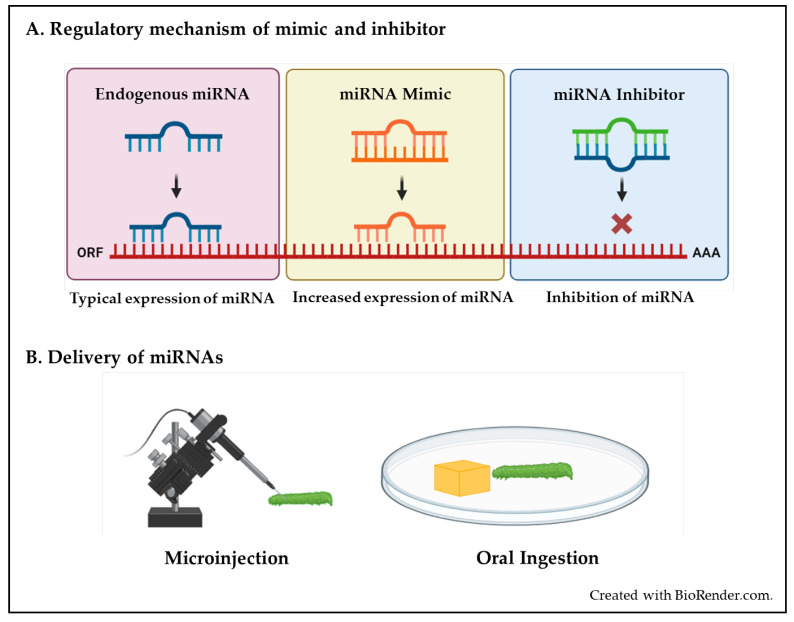
Mechanism and delivery of miRNA mimics and inhibitors in insects, (**A**) Regulatory mechanism of mimic and inhibitor, (**B**) Delivery of miRNAs.

## Data Availability

The data presented in this study is available in the article.

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
