# Peer review of "miRNA Dynamics for Pest Management: Implications in Insecticide Resistance"

_insects, 2024, doi:10.3390/insects15040238_

Round 1
Reviewer 1 Report
Comments and Suggestions for Authors
In the text, make sure to write the full scientific names of species at their first mention, even for popular species. For example, Line 140 - S. frugiperda (mentioning the full scientific name here would also benefit S. exigua and S. litura later in the text as they are all the same genus) Line 202 - C. pipiens, Line 239 - A. gossypii, and so on ....
(Suggestion) In Table 1, adding these insecticides' IRAC modes of action (MOA) would improve this table as a resource for the readers.
(Suggestion) consider moving section 7 before section 6.
(Suggestion) In Table 1, consider the reordering of the columns after "Life Stage"; ... Insecticide, MOA, Evaluation Strategy, Target Gene(s), miRNA (mediating resistance).
Line 87 - "3' UTR ...," Here would be a better place to include the meaning of UTR at first mention instead of later in Line 93.
Line 279 - Table 1, delete "Regulate resistance to" and name the column "Insecticide." Include the information about regulating resistance to the insecticides in the title or as a footnote.
Line 322 - Do the authors have additional information to add about the other approaches mentioned in Table 1? Or perhaps some insight into the application methods with additional references?
Line 335 - remove the extra period ".".
Line 367 - "MiRNA" not "miRNA"? (Line 414, similar comment)
Line 367 - add "development" after "... insecticide resistance".
Lines 367-368 - ... "providing a sustainable and innovative solution for managing pest populations," How? Not explained in the manuscript. It would be good if the authors could provide insight into recommended areas of research that align with this topic. Additionally, using these mimics or inhibitors, how would they improve or retain insecticide efficacy? Along these lines include information that highlights the potential sustainability of their use.
Lines 369-370 - "... potential of this technique to impact the mechanisms of insecticide resistance directly," Resistance mechanisms were briefly mentioned earlier in the manuscript. Consider moving this statement to the end of the section as the relation between miRNA and the mechanisms is explored in this section.
Comments on the Quality of English Language
The authors review an emerging concept that includes the use of molecular techniques in understanding insecticide resistance in many pest organisms. Using miRNA is an exciting approach to mediating resistance development and possibly mitigating the development altogether. Utilizing miRNA could also promote better monitoring techniques for resistance management. In their review, the authors highlight the current reports that use aspects of miRNA with promising results. Additionally, they discuss application approaches that have been highlighted in these reports. The review is a good article for those who utilize these techniques as it provides a comprehensive list in Table 1.
My only concerns were areas that could benefit more from tying the relation between miRNA and resistance mechanisms with actual insecticide resistance development mitigation. Although this is hinted at, a more formal discussion highlighting the sustainable benefits and further integration would have been a great addition. Similarly, this would be a good area for the authors to integrate their future implications with a more applied approach.
I recommend accepting the manuscript following some minor revisions.
Author Response
Response to reviewer’s comments
We would like to express our gratitude for the reviewer's and editor's insightful comments and suggestions, which have helped us significantly improve our manuscript's quality. We have made the changes to the manuscript based on the feedback:
Point-by-point responses to the reviewer's comments:
Notes:
- The reviewers’ comments are shown in black-colored text.
- The authors’ response to the reviewer’s comments is displayed in the blue-colored text.
- Changes made to the manuscript are indicated by red text.
Reviewer 1
- In the text, make sure to write the full scientific names of species at their first mention, even for popular species. For example, Line 140 - S. frugiperda (mentioning the full scientific name here would also benefit S. exigua and S. litura later in the text as they are all the same genus) Line 202 - C. pipiens, Line 239 - A. gossypii, and so on ....
Response: Thanks, we have corrected it throughout the manuscript.
- (Suggestion) consider moving section 7 before section 6.
Response: Thank you for your feedback. Following the approaches for identifying miRNAs involved in insecticide resistance (Section 5), we aim to emphasize the research conducted in this area to date (Section 6). Subsequently, we have described the process of functional validation of identified microRNAs (Section 7). We believe that this sequence is more effective in manuscript flow, explaining the identification of microRNAs to their functional validation.
- (Suggestion) In Table 1, adding these insecticides' IRAC modes of action (MOA) would improve this table as a resource for the readers.
- (Suggestion) In Table 1, consider the reordering of the columns after "Life Stage"; ... Insecticide, MOA, Evaluation Strategy, Target Gene(s), miRNA (mediating resistance).
Response: Thanks, we have added the MOA. Given the emphasis on the significance of miRNA and target genes in the mechanisms behind insecticide resistance, we have adjusted the order of columns in Table 1 to prioritize these elements.
- Line 87 - "3' UTR ...," Here would be a better place to include the meaning of UTR at first mention instead of later in Line 93.
Response: Thank you for pointing out. We've made the correction, moving the explanation to Line 87, where UTR is first mentioned.
- Line 279 - Table 1, delete "Regulate resistance to" and name the column "Insecticide." Include the information about regulating resistance to insecticides in the title or as a footnote.
Response: Thank you for your suggestion. We've updated the table by removing "Regulate resistance to" from the column name and now it simply reads "Insecticide." Additionally, we've incorporated the information about regulating resistance to insecticides into the title of the table for clarity and comprehensive understanding.
- Line 322 - Do the authors have additional information to add about the other approaches mentioned in Table 1? Or perhaps some insight into the application methods with additional references?
Response: We appreciate your feedback and have added more information as per your suggestion in Section 7.2 & 9.
- Line 335 - remove the extra period ".".
Response: Thank you for bringing the typographical error to our attention. We have removed the extra period as suggested.
- Line 367 - "MiRNA" not "miRNA"? (Line 414, similar comment)
Response: "miRNA" is generally used irrespective of its position. We have corrected it throughout the manuscript.
- Line 367 - add "development" after "... insecticide resistance".
Response: Thank you for your suggestion to enhance clarity. We have added "development" after "insecticide resistance" as recommended.
- Lines 367-368 - ... "providing a sustainable and innovative solution for managing pest populations," How? Not explained in the manuscript. It would be good if the authors could provide insight into recommended areas of research that align with this topic. Additionally, using these mimics or inhibitors, how would they improve or retain insecticide efficacy? Along these lines is included information that highlights the potential sustainability of their use.
Response: The application of these miRNA-based technologies has the potential to significantly improve the efficacy of insecticides by regulating the mechanisms of resistance. In addressing the query about how the use of microRNA (miRNA) mimics or inhibitors could improve or retain insecticide efficacy, and highlighting the potential sustainability of their use, we have incorporated additional information into our manuscript (Please refer to Section 9).
- Lines 369-370 - "... potential of this technique to impact the mechanisms of insecticide resistance directly," Resistance mechanisms were briefly mentioned earlier in the manuscript. Consider moving this statement to the end of the section as the relation between miRNA and the mechanisms is explored in this section.
Response: We appreciate the reviewer's insightful comment. We have revised the section.
Comments on the Quality of English Language
- The authors review an emerging concept that includes the use of molecular techniques in understanding insecticide resistance in many pest organisms. Using miRNA is an exciting approach to mediating resistance development and possibly mitigating the development altogether. Utilizing miRNA could also promote better monitoring techniques for resistance management. In their review, the authors highlight the current reports that use aspects of miRNA with promising results. Additionally, they discuss application approaches that have been highlighted in these reports. The review is a good article for those who utilize these techniques as it provides a comprehensive list in Table 1.
Response: Thank you very much for your insightful summary and comments on our review.
Lines 367-368 - ... "providing a sustainable and innovative solution for managing pest
My only concerns were areas that could benefit more from tying the relation between miRNA and resistance mechanisms with actual insecticide resistance development mitigation. Although this is hinted at, a more formal discussion highlighting the sustainable benefits and further integration would have been a great addition. Similarly, this would be a good area for the authors to integrate their future implications with a more applied approach. I recommend accepting the manuscript following some minor revisions.
Response: Thank you for your thoughtful feedback. Discussing the potential of miRNA based resistance regulation for sustainable usage, we have expanded this topic in section 9.
Reviewer 2 Report
Comments and Suggestions for Authors
I appreciate the authors comprehensive overview of the current research on the role of miRNAs in insecticide resistance. To further improve the manuscript, I would find it beneficial to include a deeper synthesis within the various sections. For example, in section 6.3 on Hemiptera, while the inclusion of multiple miRNAs linked to insecticide resistance across different species is valuable, the section would be strengthened by drawing connections between these findings. Is there a trend among different classes of insecticides? Are similar trends seen among species from different insect orders that share resistance to the same MOA? The same applies to the other order specific sections. Including broader implications of these trends for pest management strategies could provide a more interconnected analysis.
Additionally, while I noted the mention of Leptinotarsa decemlineata miRNA on lines 299-302, I feel there is a missed opportunity to delve into the role of miRNAs in coleopterans more broadly, given their major importance in pest management. Even with the lack of literature on miRNA in coleopterans compared to the other orders discussed, it would still be beneficial to dedicate a section to explore this area. This could include the information on CPB, and if other studies are limited, could perhaps speculate why research in this area is lacking, identify knowledge gaps, and suggest future research directions (ex. InsectBase includes 98 miRNAs Diabrotica virgifera including the miR-282 mentioned in the CPB study, this is a major maize pest that quickly comes to mind when considering resistance management, is anyone working on this?). Discussing these gaps could show where future research may go, and encourage further examination of these critical areas.
Section 6 (line 181) may be better titled “miRNA-mediated Insecticide Resistance in Different Arthropod Species” to reflect the inclusion of Arachnida.
Section 7 on gain and loss of function analysis lays a solid foundation for understanding these methodologies. To make this section more informative for readers, it may be beneficial to elaborate on when these methods are preferentially used. Are there certain situations in which one should choose gain of function over loss, or visa versa? Highlighting other unique aspects that differentiate the two approaches would be valuable. Explaining how they complement each other in the broader context of miRNA research would be useful for the reader, and could serve as a resource for navigating future research into the area.
In Section 9, most of the studies mentioned involve delivery via microinjection. It may be beneficial to note that this preliminary work is foundational, while drawing more attention to the methodologies that are more relevant to field conditions. Specifically, the study in P. xylostella (line 394) which incorporates miRNA mimics in the diet, represents an advancement toward practical application for pest management. While both are critical to discuss, in the broader context of insecticide resistance, bringing attention to how dietary miRNA mimics and similar technologies could move this toward pest management would be useful.
Overall, this review has potential to be a valuable contribution to the field. I would recommend a more thorough analysis and interpretation of the studies and techniques discussed throughout the manuscript. Delving deeper into the implications and significance of these findings would provide readers with a better understanding of the research currently available on this topic, and provide valuable future directions. Expanding on practical applications and limitations of the research discussed would strengthen this review.
Comments on the Quality of English LanguageOverall, I found the language and grammar to be appropriate, and see only minor revisions being necessary. Some language could be a bit more informative (in some situations it is stated an miRNA “alters” resistance—while appropriate in some instances, in others it may be better to specifically state if it increases resistance, or if it reduces it).
Author Response
Response to reviewer’s comments
We would like to express our gratitude for the reviewer's and editor's insightful comments and suggestions, which have helped us significantly improve our manuscript's quality. We have made the changes to the manuscript based on the feedback:
Point-by-point responses to the reviewer's comments:
Notes:
- The reviewers’ comments are shown in black-colored text.
- The authors’ response to the reviewer’s comments is displayed in the blue-colored text.
- Changes made to the manuscript are indicated by red text.
Reviewer 2
- I appreciate the authors comprehensive overview of the current research on the role of miRNAs in insecticide resistance. To further improve the manuscript, I would find it beneficial to include a deeper synthesis within the various sections. For example, in section 6.3 on Hemiptera, while the inclusion of multiple miRNAs linked to insecticide resistance across different species is valuable, the section would be strengthened by drawing connections between these findings. Is there a trend among different classes of insecticides? Are similar trends seen among species from different insect orders that share resistance to the same MOA? The same applies to the other order specific sections. Including broader implications of these trends for pest management strategies could provide a more interconnected analysis.
Response: There is emerging evidence that miRNAs can indeed influence insecticide resistance, potentially across different classes of insecticides. However, the specificity of miRNA-mediated resistance to various insecticide classes can vary and remains a subject of ongoing research. Further, resistance mechanisms, including those potentially regulated by miRNAs, can be conserved across species sharing resistance to the same mode of action (MOA) of insecticides. However, the extent to which similar miRNA profiles confer resistance in species from different orders is an area that warrants further investigation.
Additionally, the context-dependent nature of miRNA action, wherein a single miRNA can target multiple genes and a single gene can be targeted by multiple miRNAs and the conserved nature of miRNAs across insect species, adds a layer of complexity to deciphering these interactions. We have incorporated detailed information in Section 4 (Line 148-171).
- Additionally, while I noted the mention of Leptinotarsa decemlineata miRNA on lines 299-302, I feel there is a missed opportunity to delve into the role of miRNAs in coleopterans more broadly, given their major importance in pest management. Even with the lack of literature on miRNA in coleopterans compared to the other orders discussed, it would still be beneficial to dedicate a section to explore this area. This could include the information on CPB, and if other studies are limited, could perhaps speculate why research in this area is lacking, identify knowledge gaps, and suggest future research directions (ex. InsectBase includes 98 miRNAs Diabrotica virgiferaincluding the miR-282 mentioned in the CPB study, this is a major maize pest that quickly comes to mind when considering resistance management, is anyone working on this?). Discussing these gaps could show where future research may go and encourage further examination of these critical areas.
Response: Your observation highlights an important gap in the review of miRNAs in Coleoptera and their implications for pest management. We have added section 6.4 that includes reference of CPB. Although numerous miRNAs have been identified, there is lack of research on miRNAs associated with insecticide resistance within Coleoptera.
- Section 6 (line 181) may be better titled “miRNA-mediated Insecticide Resistance in Different Arthropod Species” to reflect the inclusion of Arachnida.
Response: We appreciate your attention to the details and revised the section title to "miRNA-mediated Insecticide Resistance in Different Arthropod Species."
- Section 7 on gain and loss of function analysis lays a solid foundation for understanding these methodologies. To make this section more informative for readers, it may be beneficial to elaborate on when these methods are preferentially used. Are there certain situations in which one should choose gain of function over loss, or vice versa? Highlighting other unique aspects that differentiate the two approaches would be valuable. Explaining how they complement each other in the broader context of miRNA research would be useful for the reader and could serve as a resource for navigating future research into the area.
Response: Thank you for your insightful feedback. Gain and loss of function experiments are not mutually exclusive and often serve complementary roles in miRNA research. By employing both approaches, researchers can achieve a comprehensive understanding of miRNA functions. For example, a miRNA identified as critical in gain-of-function experiments can be further validated through loss-of-function experiments, and vice versa. This dual approach allows for the confirmation of miRNA target relationships, elucidates miRNA regulatory networks, and enhances the understanding of miRNA roles in complex biological processes like insecticide resistance. The choice between gain and loss of function may also be dictated by the specific biological question at hand. For instance, if the research aims to uncover the function of a miRNA in resistance mechanisms, a loss-of-function approach using antagomirs might be preferred to observe effects of reduced miRNA activity on resistance phenotype. Conversely, if the aim is to identify new functions or enhance known functions of a miRNA, gain-of-function experiments with miRNA mimics can provide valuable insights. We have incorporated additional details into the manuscript Section 7 (Line 339 to 349, 356-358.).
- In Section 9, most of the studies mentioned involve delivery via microinjection. It may be beneficial to note that this preliminary work is foundational, while drawing more attention to the methodologies that are more relevant to field conditions. Specifically, the study in xylostella (line 394) which incorporates miRNA mimics in the diet, represents an advancement toward practical application for pest management. While both are critical to discuss, in the broader context of insecticide resistance, bringing attention to how dietary miRNA mimics and similar technologies could move this toward pest management would be useful.
Response: Thank you for the recommendation to improve the manuscript's focus. Indeed, both microinjection and dietary delivery of miRNA mimics play crucial roles in the exploration of miRNA-mediated pest control strategies. Yet, the application of miRNA mimics through diet emerges as a more feasible and scalable option for field applications. However, field application has been proven effective in a few studies among herbivorous and piercing-sucking insects (Line: 506-508).
In context of lab experiments, miRNA efficiency is high if it is delivered by microinjection but comparably low if it is delivered by diet feeding. The lower efficiency of miRNA delivery by diet feeding appears to be mainly due to degradation of miRNA mimics in the insect gut. The insect gut environment, characterized by its pH, enzymatic activity, and rapid digestion processes, can potentially degrade, or inactivate miRNA mimics before they can exert their intended regulatory effects on gene expression. Nonetheless, continued research and development are necessary to optimize delivery methods, enhance the stability and efficacy of miRNA treatments, and ensure their practicality and safety for large-scale use. We have briefly incorporated and discussed these considerations in Section 8.2 (Line 455-457) & and elaborated on the practical application in Section 9.
- Overall, this review has potential to be a valuable contribution to the field. I would recommend a more thorough analysis and interpretation of the studies and techniques discussed throughout the manuscript. Delving deeper into the implications and significance of these findings would provide readers with a better understanding of the research currently available on this topic and provide valuable future directions. Expanding on practical applications and limitations of the research discussed would strengthen this review.
Response: We greatly appreciate your constructive feedback, which will undoubtedly strengthen the quality and impact of our review. Your suggestion to expand on the practical applications and limitations of the research discussed is particularly valuable. We have revised our manuscript to incorporate a more detailed analysis and discussion of the studies, their implications, and the potential for field applications in Section 9.
- Comments on the Quality of English Language
Overall, I found the language and grammar to be appropriate, and see only minor revisions being necessary. Some language could be a bit more informative (in some situations it is stated a miRNA “alters” resistance—while appropriate in some instances, in others it may be better to specifically state if it increases resistance, or if it reduces it).
Response: Thank you for the valuable suggestion. We have carefully revised the relevant sections of our manuscript.
Round 2
Reviewer 2 Report
Comments and Suggestions for Authors
The revisions made have greatly improved the clarity and depth of the review. Having reviewed the authors response to my comments, I am comfortable recommending the manuscript for acceptance in Insects.
Comments on the Quality of English LanguageAfter looking over the manuscript the authors’ may find a few word choices or edits to make, but overall nothing major stood out to me.